# Wetting Behavior and Tribological Properties of Polymer Brushes on Laser-Textured Surface

**DOI:** 10.3390/polym11060981

**Published:** 2019-06-04

**Authors:** Ming-xue Shen, Zhao-xiang Zhang, Jin-tao Yang, Guang-yao Xiong

**Affiliations:** 1College of Materials Science & Engineering, East China Jiaotong University, Nanchang 330013, China; 2College of Materials Science & Engineering, Zhejiang University of Technology, Hangzhou 310032, China; 3State Key Laboratory of Tribology, Tsinghua University, Beijing 100084, China

**Keywords:** polymer brushes, textured surface, wettability, load-bearing capacity, friction reduction and anti-wear

## Abstract

Polymer brush layers can act as effective lubricants owing to their low friction and good controllability. However, their application to the field of tribology is limited by their poor wear resistance. This study proposes a strategy combining grafting and surface texturing to extend the service life of polymer brushes. Surface microstructure and chemical composition were measured through scanning electron microscopy (SEM), atomic force microscopy (AFM), and X-ray photoelectron spectroscopy (XPS). Water contact angles were measured to evaluate the surface wettability of the grafted silicon-based surface texture. Results showed the distinct synergistic effect between polymer brushes and laser surface texturing (LST). The prepared polymer brushes on textured surface can be a powerful mechanism for friction reduction properties, which benefit from their strong hydration effect on the lubrication liquid and promote the formation of a local lubricating film. Moreover, the wear life of polymer brushes can be immensely extended, as micro-dimples on the textured surface can effectively protect the polymer brushes. This study presents a method to enhance the load-bearing capacity and wear resistance of the grafted surface of polymer brushes.

## 1. Introduction

To obtain low friction coefficients on substrate surfaces, the use of polymer brushes prepared from different monomers as a protective coating was extensively investigated in recent years [1,2]. Grafted surfaces were observed to exhibit super lubrication in specific environments with various neutral, cationic, anionic, and zwitterionic polymer brushes [3,4]. Thus, brushes with different monomers can be used as symptomatic remedies for different lubrication conditions, such as water, oil, salt solution, acid solution, and high-temperature lubrication [5,6,7,8]. Efficient lubrication of the polymer brush can be attributed to the highly stretched conformation of polymer chains in good solvents due to solvation, and the generation of osmotic repulsive force is opposite to the applied load, which can effectively prevent the approach of polymer chains and avoid entanglement between polymer chains. Thus, a flowing lubrication hydration layer easily forms between frictional interfaces, resulting in an ultra-low friction coefficients when the interface is sheared [9,10,11]. 

Although polymer brushes can achieve ultra-low friction coefficients (usually, friction coefficients are in the range of 0.001–0.008), common problems such as low load-bearing capacity and easy wearing still exist, which immensely limit their application in the industry [12]. On this basis, a cross-linked strategy for polymer brushes was proposed to enhance load bearing and wear resistance of their surface by our research group [13]. However, the friction coefficient of the surface remarkably increased. Methods for extending the service life of a polymer layer-grafted surface became an insurmountable problem in the research of low-friction polymer grafting.

Laser surface texturing (LST) was recognized as an effective means of surface engineering to improve the tribological properties of sliding surfaces [14,15,16,17,18]. A combination of LST and solid lubricants was utilized to achieve friction reduction and extension on wear resistance life, as micro-dimples can reduce the contact area and protect lubricants in the pit from wear [19,20,21,22]. For example, Ripoll et al. [21] and Hu et al. [22] produced micro-dimple patterns on a Ti_6_Al_4_V alloy and coated MoS_2_ film on textured surfaces (TS) to improve their tribological properties, and they observed a remarkable reduction in friction coefficient and an improvement in the lifetime of the MoS_2_ film. Voevodin et al. [23] fabricated micro-dimples on coated steel surfaces to improve the tribological properties of TiCN coating, and their results indicated that the lifetime of solid lubricants on dimpled surfaces was an order of magnitude longer than that of the unmodified TiCN coating surface.

A strategy combining grafting and texturing is proposed in the present work to extend the service life of the grafted surface of polymer brushes. Previous research showed that textured surfaces can improve load-carrying capacity, grafted polymer brushes can improve lubricity, and surface micro-dimples can protect polymer chains from wear. In this study, the tribological properties, combined with the textured surface and the polymer brushes, were discussed to determine their synergistic effect. In particular, a monocrystalline silicon surface was initially laser-textured first, followed by grafting polymer brushes on the textured surface. Then, the wettability and tribological properties of the grafted polymer brushes on smooth and textured silicon surfaces were compared and analyzed. To provide new insights and reference for prolonging the wear life of grafted surfaces of the polymer brush, the wear mechanism of grafted polymer brushes on the matrix surface was evaluated through a detailed analysis of their surface morphology and composition.

## 2. Experimental Procedure

### 2.1. Materials

The chemicals 4-vinylbenzyl chloride (90%), dimethylamine solution (40 wt.% in H_2_O), 1,3-propanesultone (98%), and copper(I) bromide (98%) were purchased from Sigma-Aldrich (Shanghai). Tris[2-(dimethylamino)ethyl]amine (Me6TREN; 99%) was purchased from J&K Chemical Ltd. (Beijing, China). The initiator for grafting polymer brushes on silica wafer, 3-(2-bromoisobutyramido)propyl(trimethoxy)silane, was purchased from Gelest, Inc. (Morrisville, PA, USA). The water used in experiments was purified using a Millipore water purification system with a maximum resistivity of 18.0 MΩ∙cm. The N-(4-vinylbenzyl)-N,N-dialkylamine (SVBA) monomer was synthesized and purified using a previously published method [24]. All other reagents and solvents were commercially obtained as extra-pure grade and used as received.

The matrix materials used in experiments were flat and textured single-crystal silicon wafers. The texture density of lubricated components should be between 5% and 15% and should not exceed 20% to obtain good lubrication performance [14,15], Hu et al. observed that the combination of solid lubricants and textured surfaces with 23% dimple density yielded a low friction coefficient and obtained a long wear life by increasing the dimple density [22]. We chose 20% density as the texture parameter based on these aspects. As shown in Figure 1, the micro-dimples were fabricated through laser micromachining, and the depth of the dimple was changed by adjusting the output power of the laser marker. The laser was emitted at the fundamental wavelength of 1064 nm, with a 20-kHz pulse repetition rate; the output power was set to 0.3 W, and marking velocity was set to 100 mm/s. The dimple radius was approximately *r* = 25 μm, and area densities were around 20%. Figure 1c shows the schematic of the textured surface, and related dimension parameters are marked in the diagram for subsequent calculation.

### 2.2. Preparation of Polymer Brushes

Zwitterionic polymer brushes were synthesized through the surface-initiated atom transfer radical polymerization (SI-ATRP) method (Figure 2). A few sheets of wafers (25 mm × 15 mm) were ultrasonically cleaned in ethanol and deionized water, and then were placed into a fresh piranha solution (H_2_SO_4_:H_2_O_2_ = 3:1) at 120 °C for about 0.5 h. After the samples were repeatedly washed with deionized water and dried with N_2_ flow, the wafer was subsequently treated with plasma (CORONA Lab. CTP-2000, Nanjing, China) for about 2 min. The cleaned samples were immediately immersed into a 1 mM dehydrate toluene solution containing 3-(trimethoxysilypropyl)-2-bromo-2-methylpropionate for 12 h at room temperature. The initiator-grafted surface was sequentially washed with toluene, ethanol, and water, and then dried under a stream of nitrogen.

The SVBA monomer (1.96 mmol) and Me6TREN (0.14 mmol) were dissolved in water (2.5 mL), and the mixture was degassed by flowing a stream of nitrogen for 20 min. Then, 2,2,2-trifluoroethanol was also degassed in the same manner. CuBr (15.7 mg, 0.11 mmol) and the initiator-coated textured and flat silicon surface were placed in the same reaction tube back to back. The tube was immediately evacuated and backfilled with nitrogen three times to remove oxygen. The degassed 2,2,2-trifluoroethanol (2.5 mL) and water solution containing the monomer and ligand were added using a syringe to the reaction tube under nitrogen protection. The tube was then subjected to two evacuation and nitrogen purging cycles and kept at room temperature for the SI-ATRP reaction. After the prespecified reaction time (20 h and 48 h), the solution was exposed to air to terminate the reaction. The grafted wafer was collected, washed with 2,2,2-trifluoroethanol and saturated NaCl solution to remove the free polymer absorbed on its surface, and dried through nitrogen flow. The samples were stored in air in room conditions until the surface analysis or the friction test.

### 2.3. Test and Characterization

The salt solution contact angles of grafted polymer brush samples were recorded with an OCA 15EC video-based optical contact-angle measuring system (Eastern-Dataphy Instruments Co., Ltd., Beijing, China). A 5-μL droplet of salt solution was dropped on the dry surface with a microsyringe. The data presented were the average of five independent measurements on different positions.

Friction and wear tests were conducted with a ball-on-plate type UMT-3 friction tester (Bruker Company, Hamburg, Germany). During sliding, poly(dimethylsiloxane) (PDMS) hemispheres 6 mm in diameter were used as the counterparts. The displacement amplitude was 5 mm and the fixed reciprocating frequency was 2 Hz. A 2-N normal load was applied in this test, and each friction test was performed under saturated NaCl solution lubrication conditions in an ambient environment. All friction tests were repeated at least three times for each test parameter. To evaluate the friction reduction capability of polymer brushes with the most stringent criteria, we averaged the maximum friction values of the two directions in each reciprocating cycle during data collection.

The morphology before and after the friction test was analyzed though scanning electron microscopy (SEM, JSM-6610, JEOL, Tokyo, Japan) with an accelerating voltage of 15 kV. The detailed microstructure and the roughness of specimens were evaluated through atomic force microscopy (AFM) with an SPI3800N instrument (Seiko, Tokyo, Japan) under tapping mode. A quick scan of a 200 μm × 200 μm area was initially collected, followed by small scan sizes in the desired regions of interest.

The chemical composition changes on the surfaces before and after friction test were detected through X-ray photoelectron spectroscopy (XPS, ESCALAB 250Xi, Thermo Fisher Scientific, Carlsbad, NM, USA); the binding energy (BE) scale was calibrated by C_1*s*_ of the hydrocarbon peak at 284.5 eV, and a Shirley-type background was chosen for the XPS fitting.

## 3. Results and Discussion

### 3.1. Surface Composition and Morphology Analysis of PolySVBA Brushes

To confirm the modified surface, XPS was performed to analyze and compare the surface compositions of initiator-modified and polymer brush-grafted surfaces. As shown in Figure 3, a Br_3*d*_ characteristic peak near the binding energy (BE) of 69.8 eV and a C_1*s*_ peak at 284.5 eV (Figure 3a) were observed, indicating that the initiator was successfully grafted to the silicon surface as the original surface only contained silicon, and bromine is a unique component of the 3-(trimethoxysilypropyl)-2-bromo-2-methylpropionate initiator. For the sample after polymerization (Figure 3b), the C/O ratio remarkably increased, indicating that the carbon chain length on the surface remarkably increased. In addition, the high-resolution C_1*s*_ peak on the surface of the grafted polySVBA (PSVBA) sample could be decomposed into two peaks at 284.5 eV and 286.1 eV (Figure 3c), which corresponded to C–C/C–H/C–S/C=C and C–N/C–O, due to the benzyl group and sulfobetaine motif from the PSVBA brushes. The N_1*s*_ peak could be decomposed into two peaks at 400.1 eV and 402.3 eV (Figure 3d), which were attributed to N–C and –N^+^(CH_3_)_2_–, respectively. Particularly, the S_2p_ peak of sulfonic acid group appeared on the grafted surface at 167.9 eV (Figure 3e). Detailed analysis of the nitrogen and sulfur confirmed the existence of SVBA on the surface, and the results were consistent with findings in literature [25,26]. Thus, PSVBA polymer brushes were successfully grafted onto the sample surface.

The morphology of PSVBA brushes grafted with flat silicon wafers was characterized through AFM (Figure 4). A smooth grafted surface was observed after 20 h of polymerization, and its root-mean-square (RMS) roughness was approximately 6.05 nm (Figure 4a). When grafting time was increased to 48 h, the surface became rough, considerable particle peaks (particle-like topography due to the collapse of attached polymer chains under “dry” conditions [27]) were distributed on the surface, and RMS increased to 10.65 nm (Figure 4b), similar to the findings of a previous report [28]. The above characteristics were observed on flat and textured surfaces, except in the micro-dimple region.

The micro-dimples cannot be accurately characterized through AFM due to the serious ablation and rough morphology caused by laser processing. As shown in Figure 5, SEM observation showed a small difference between the morphologies of the 20-h grafted surface and ungrafted surface (see Figure 5a), and this result may be due to the nanosized grafting thickness of the polymer, resulting in the overall morphology of the substrate being unaffected at the micron scale. When the reaction time increased to 48 h, more aggregation of colloidal polymers was observed around the circular pores than in the plane region (see Figure 5c), and several micro-dimples were filled with these polymers. This condition may be related to the difficulty of cleaning the polymer layers that were physically adsorbed by micro-dimples. However, the polymer layer on the surface grafted for 48 h was thicker than that grafted for 20 h, whether in the plane or texture area.

### 3.2. Analysis of Surface Wetting Behavior

The wetting phenomenon plays an important role in surface lubrication. After successful grafting of PSVBA brushes on flat and textured silicon surfaces, static contact angles of the surfaces in saturated NaCl solution with different modifications were measured. As shown in Figure 6, the contact angle of the original silicon wafer surface of the flat sample was approximately 88.5°, which reduced to 76.4° after coating of the self-assembled monolayer initiator, and 37.9° after grafting of the PSVBA brushes via SI-ATRP, indicating the high hydration of PSVBA brushes in the salt solution. Polymers exert an “anti-polyelectrolyte effect” with salt-responsive behaviors similar to PSVBA brushes, in which the electrostatic inter/intrachain dipole–dipole interaction is destroyed under high concentration of salt solution, and chain–chain interactions are reduced, promoting polymer–water interactions at the interface, which results in low contact angles [29,30], and conformational changes in polymers from a collapsed to an extended conformation as confirmed recently [31,32]. For the textured silicon wafer surface, the contact angle decreased from 109.1° to 100° and 39.0°. Although the contact angle of the original textured surface was significantly higher than the flat silicon surface, the contact angle of the PSVBA grafted textured surface was close to that on the grafted flat surface (FS). The results show that the grafted PSVBA brushes can effectively improve the wettability of flat and textured silicon surfaces, and their wetting effects are equivalent.

The Wenzel model and Cassie–Baxter model are often used to analyze the wetting behavior of rough surfaces [33]. The Wenzel model believes that the droplets completely wet all contact surfaces, making the hydrophilic surface more hydrophilic and hydrophobic surfaces become more hydrophobic [34]. However, the Cassie–Baxter model considers that the gases in the textured pores will support the droplets and make the surface more hydrophobic [35]. In order to deeply analyze the wetting behavior of the surface with different modifications, the predicted values of both theoretical models were compared with the experimental values in this work.

The *r* is the ratio of the solid–liquid actual contact area over the projected surface area in the Wenzel model, as shown Figure 1, where the definition of *r* in this work is as follows:(1)$$r=\frac{c^{2}-\pi r^{2}+\pi r\cdot \sqrt{h^{2}+r^{2}}}{c^{2}}.$$

In the Cassie–Baxter model, *f* is the ratio of the solid–liquid contact area over the projected surface area, and its definition in this work is the following equation:(2)$$f=\frac{c^{2}-\pi r^{2}}{c^{2}}.$$

According to the above formulae, the theoretical values of each model can be calculated. The specific data are listed in Table 1, and the theoretical values can be compared with experimental measurement values. For the initial surface, the surface wetting behavior after texturing was obviously in accordance with the Cassie–Baxter model. The air inside the micro-dimples plays an important role to support the droplets on the surface. When the initiator layer was grafted on the surface, although the contact angle of the surface decreased and the wettability improved, it can be found that the surface wetting state still belonged to the Cassie–Baxter state. When the PSVBA brushes were grafted on the textured surface, and the contact angle $\theta_{w}<\theta_{textured}<\theta_{CB}$, it was obvious that the wetting state of the surface changed in such a situation.

To analyze the surface wetting behavior in detail, the contact angle of PSVBA brushes grafted on the textured surface was measured with the increase in time (Figure 7). Figure 7a shows the shape of the droplet on the surface 10 s after dropping. The gas in the textured pore supported the droplet and formed a light–dark fringe inside the droplet. The contact angle was approximately 66.6°, which is close to the theoretical calculation results of the Cassie–Baxter model, indicating that the surface wetting state is the Cassie–Baxter state. The liquid droplet gradually filled the micro-dimples of the textured substrate, accompanied with a decrease in the apparent contact angle due to the strong adsorption effect of the PSVBA brushes on the salt solution. The bubbles in the droplet gradually moved up (see Figure 7a–c), and the effect of gases in the textured pores on surface wetting behavior gradually weakened, whereby the wetting behavior changed from the Cassie–Baxter state to the Wenzel state (see Table 1 and Figure 7d). This condition is similar to the change in solid/liquid contact mode from the Cassie–Baxter state to the Wenzel state when the spherical liquid droplet is pressed physically, or the energy barrier is overcome by external forces [36,37]. In general, it can be concluded that the textured surface can make the silicon surface more hydrophobic; however, the method of grafting polymer brushes on the surface is an effective way to change the surface wettability, and is less affected by the substrate surface topography.

### 3.3. Analysis of Friction Coefficient and Wear Resistance

Figure 8 shows the average friction coefficients at the initial stable stage for different modifications. It can be seen that the friction coefficient of PSVBA brushes grafted on flat surfaces is less than 0.03, which is approximately 1/50 of the original surface. Meanwhile, the friction coefficient of the textured surface reached 0.964, which is approximately 32% lower than that of the flat surface. The surface texture anti-friction was widely confirmed by various reports [38]. This condition is attributed to surface micro-pits that can reduce the contact area and effectively improve the fluid-bearing capacity on the surface, thus reducing friction and wear under lubrication. In addition, grafting PSVBA brushes can immensely reduce the friction coefficient of the textured surface, and the friction reduction effect is higher compared with that on the flat silicon surface. Significant anti-friction effects of the grafted surface of polymer brushes were reported by a number of researchers [4,39]. Other studies showed that the excellent lubrication performance of PSVBA brushes is due to their ability to hydrate under high concentrations of salt solution (Figure 6). This condition results in a highly stretched conformation of polymer chains in the interface and the generation of an osmotic pressure repulsive force from polymer chains opposite to the applied load, which can effectively improve the load-carrying capacity of the surface and promote the formation of a thin fluid lubrication film, resulting in a low surface friction coefficient.

Although polymer brushes feature excellent lubrication effects, the wear life of their surface is generally insufficient, which limits their application in the industry. Similarly, as shown in Figure 9a, the FS-PSVBA-20 h (PSVBA grafted on flat surface for 48 h) surface exhibited excellent lubricating effects in this study, but its surface showed poor wear resistance, as its wear life could only maintain 2000–3000 test cycles. When the micro-dimples were fabricated on the surface, a low friction coefficient could be maintained to approximately 5000 friction cycles, indicating that the dimples can effectively prolong the wear life of polymer brushes. This condition may be explained by the combined elastohydrodynamic lubrication (EHL) theory. The EHL theory states that the load capacity is proportional to (*L*/*h*)^2^, and the friction force is proportional to (*L*/*h*), where *L* refers to the bearing length, and *h* denotes the minimum film thickness. This condition indicates that, when EHL can be established at the polymer brush-grafted surface, then the sum of the dynamic pressure-bearing capacity of the fluid film and the osmotic pressure repulsion force of the polymer brush chains must be able to offset the external load. FS-PSVBA-20 h can achieve such EHL conditions at the initial stage. However, the continuous wear of polymer chains caused the decrease in osmotic pressure repulsion force, and the fluid film could not maintain sufficient support force with the progress of the friction test. Therefore, the film thickness decreased, the friction pair gradually contracted, the conformation of polymer brush cannot be fully extended along the load direction, and the shear resistance of the solution increased, resulting in increased friction force, breakage of polymer chains, and surface failure. For the TS-PSVBA-20 h (PSVBA grafted on textured surface for 48 h) surface, although the polymer brushes could stretch conformation in a good solution, they remained shorter compared with the micro-dimples in the depth range of 20–30 microns. Thus, the polymer chains at the bottom of micro-dimples cannot affect the lubrication behavior of the surface, and a small distance between friction pairs and a high friction coefficient will be obtained. The friction coefficient sharply rose again when the polymer chains with anti-friction effect were completely worn out.

The anti-wear effect of TS-PSVBA-20 h is unsuitable for industrial applications. On this basis, the reaction time was extended to 48 h to improve the thickness of PSVBA brushes. As referred to in the literature [13,30], with the increase in thickness of the grafted layer, the surface became rough, and the friction coefficient increased. This condition can be explained as follows: the high thickness of polymer brushes resulted in the significant decrease in load-carrying capacity of the fluid film, whereas the friction pairs must be close to each other, causing difficulty in full extension of polymer chains along the load direction and serious wearing, resulting in a high friction coefficient during the friction test, as shown in Figure 9b. The friction coefficient of the PS-PSVBA-48 h surface rapidly increased after 12,000 cycles of friction test, indicating that the grafted layer lost its friction-reducing function. Although the friction coefficient of the 48-h grafted textured surface was increased to 0.5 (one-third of the original flat silicon surface), the friction test could be stably operated about 35,000 cycles without significant fluctuations, as the polymer chains in the dimples formed no direct contact with the friction pair, and could affect the lubrication effect on the surface for a long time. The results showed that the textured surface can significantly improve the wear resistance of the polymer brushes, and the thickness of the grafted layer should be suitable for the depth of texture.

### 3.4. Wear Mechanism

To analyze the related phenomena during the friction test, the wear morphology under different modified surfaces was observed through SEM, as shown in Figure 10. For the polymer brushes grafted on the flat surface for 20 h, the worn surface was smooth, and the signs of grafted polymer brushes could not be distinguished (Figure 10a). XPS detection showed that the C/O ratio on the worn surface at this time was significantly lower than that on the non-worn surface (compared with Figure 3b and Figure 10a), indicating that the grafted polymer chains were severely broken, which indirectly confirmed that the wear of grafted layer primarily caused the increase in friction coefficient. The wear surface of polymer brushes grafted for 48 h could easily distinguish the wear and the non-wear areas, and the polymer brush film was completely worn after a long-period friction test (Figure 10b). However, the worn morphology of the texture surface immensely changed with different grafting times. Although the surface friction coefficient of polymer brushes grafted for 20 h was smaller than that for 48 h, a lot of wear debris was generated and removed from friction interfaces with the increase of friction cycles, which gradually aggregated on both sides of wear scars along the friction direction (see Figure 10d). This debris probably resulted from the aggregation of fractured polymer chains during the repeated friction test. “Brush” conformation was lost with the agglomeration of polymer chains, that is, the excellent lubrication effect of polymer brushes was lost, and friction coefficient rapidly increased (see Figure 9a). By contrast, no agglomeration occurred on the surface grafted for 48 h, and no significant difference was observed in the surface morphologies of micro-dimples before and after wearing, as indicated in the SEM photographs. This is the main reason why the surface could maintain long-term stability during a prolonged friction test.

To further analyze the failure mechanism of polymer brush-grafted surfaces during wear on the ultrastructure, the AFM morphology of 48-h PSVBA brushes grafted on flat silicon wafers was determined at different wear stages, as illustrated in Figure 11. The wear of polymer brushes on the surface was divided into four stages. In the initial wear stage (e.g., *N* = 1000), the surface of the polymer brushes was covered with numerous micro-grooves (see Figure 11a), which may be due to ploughing by PDMS pairs, and the wear mechanism was mainly a two-body abrasive wear. In the fluctuation stage (e.g., *N* = 4000), the micro-grooves on the grafted surface were obviously deepened, and the local area showed evident “negative wear” morphology (see Figure 11b), which may be due to the pulling effect on the local polymer chain in the wear area during the two-body abrasive wear process, resulting in increased height of polymer chains in this area. Therefore, the friction coefficient remarkably fluctuated at this stage, which led to a sharp increase in surface roughness (RMS = 23.35 nm). Then, it reached the stable wear stage (e.g., *N* = 8000), as shown in Figure 11c, with the continuation of the wear process; the friction coefficient maintained a stable value, the whole wear morphology was relatively flat, and the local area of “negative wear” morphology disappeared, while the characteristics of polymer chain fracture could be seen clearly in the local position of the wear surface. The polymer chains on the grafted surface were gradually cut off and the wear mechanism was changed into fatigue wear. Finally, as shown in Figure 11d, the failure stage (e.g., *N* = 13450) was followed by the continuous fracture of the polymer chains; the thickness of the grafted layer gradually decreased, the surface of the matrix began to be exposed, the micro-grooves almost disappeared, the friction coefficient rose sharply in the meantime, the roughness of the worn surface was close to the original substrate surface after the 13,450-cycle friction test, and the grafted polymer brushes were almost completely worn away. Of course, the task of the polymer brushes was terminated after the final stage.

Figure 12 shows a schematic of the friction mechanism of polymer brushes grafted on flat and textured silicon surfaces on the basis of the obtained experimental results. The polymer chains were evenly stretched along the height direction and arranged in the same direction as each other, when the polymer brushes were grafted on the flat surface, gradually forming a uniform hydration layer, and resulting in a low friction on the grafted surface. However, the polymer chains were easily destroyed under the alternating shear effect, as only an extremely thin hydration layer separated the friction interfaces and carried the normal load. Thus, the wear life of the surface was generally insufficient. The stretching of polymer chains in good solvents relatively differed from that in the grafted flat surface, when the polymer brushes were grafted on the texture surface; that is, the arrangement of polymer chains changed with the substrate curvature [40], resulting in poor flow of hydration layer on textured surface compared with that on the flat substrate surface. In addition, given the numerous pores on the surface, the number of polymer chains on the textured surface in direct contact with the friction pair will be reduced. Thus, the osmotic pressure repulsive force acting on the friction pair was weakened. The combination of the two factors led to a higher friction coefficient on the grafted texture surface than on the grafted flat surface in the initial stage of the friction test. On the other hand, although most of the polymer brushes in the micro-dimples showed no direct contact with the friction pair, they still presented good wettability (see Figure 6 and Figure 7). Thus, the polymer brushes can effectively lock the lubricant and store it in the micro-dimples to facilitate a hydrodynamic lubrication film during the friction process. However, the polymer chains should be longer than the depth of micro-dimples, as short polymer chains at the bottom cannot affect the lubrication behavior on the surface (see Figure 7b,c). Once the polymer chains in the pit can affect the surface lubrication effect, the formation of a hydrodynamic lubrication film can prevent breakage of the polymer chains in the non-textured region (i.e., outside the micro-dimples). Moreover, the polymer chains in micro-dimples exhibited no direct contact with the tribo-pairs, which can effectively protect the polymer chains from wearing. Thus, the polymer chains can influence the lubrication effect between friction interfaces for a long period. Therefore, this method can remarkably extend the service life of polymer brushes (see Figure 10).

## 4. Conclusions

The combined effects of PSVBA polymer brushes as a lubricating film and LST on improving the wettability, reducing friction, and extending the service lifespan of polymer brushes were investigated. The following conclusions were drawn:The wettability of substrate surface can be significantly improved by grafting PSVBA brushes, gradually changing from the Cassie–Baxter state to the Wenzel state by grafting polymer brushes, such that the texture causes no effect on the grafted surface wettability.For the grafted smooth surface, polymer brushes easily form a hydrated layer and exhibit ultra-low friction behavior. However, the alternating shearing action can easily break the polymer chains and, thus, destroy the lubrication film, and the short wear resistance life of polymer brushes cannot be avoided.Although the friction coefficient of the grafted textured surface slightly increased, the friction test on the surface could be stably operated for 35,000 cycles without significant fluctuations, as the micro-pits on the textured surface can protect polymer brushes from wear. The effect of the micro-texture can immensely prolong the service life of polymer brushes and expand their tribological applications.

## Figures and Tables

**Figure 1 polymers-11-00981-f001:**
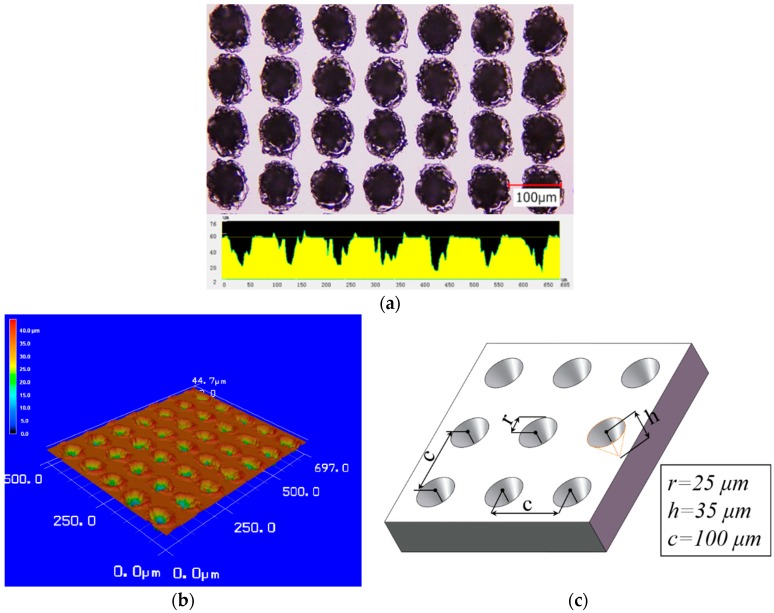
(**a**) Two-dimensional (2D) and (**b**) three-dimensional (3D) optical micrographs of the textured surface; (**c**) schematic of 3D topography.

**Figure 2 polymers-11-00981-f002:**
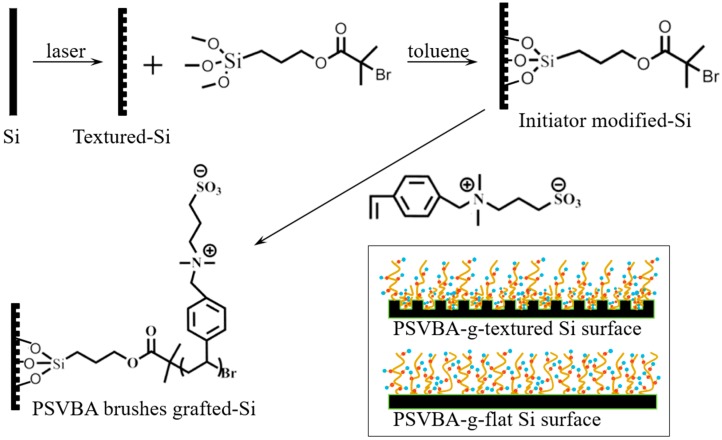
Schematic of SI-ATRP process for grafting polySVBA onto a silicon wafer.

**Figure 3 polymers-11-00981-f003:**
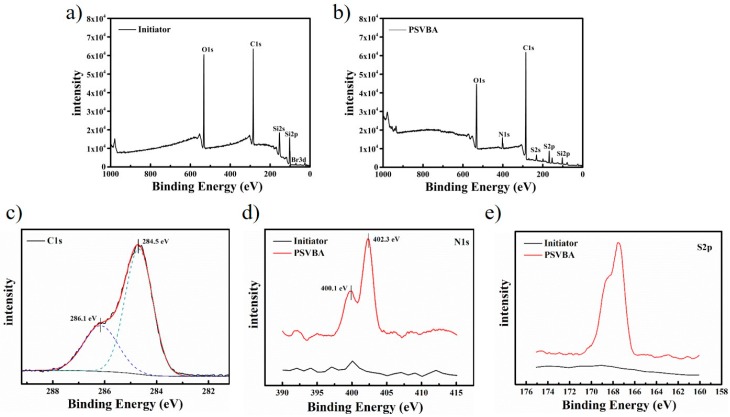
X-ray photoelectron spectroscopy (XPS) survey spectra of silicon surfaces grafted with (**a**) initiators and (**b**) polySVBA (PSVBA) brushes; (**c**) C_1*s*_, (**d**) N_1*s*_, and (**e**) S_2*p*_ high-resolution survey scans.

**Figure 4 polymers-11-00981-f004:**
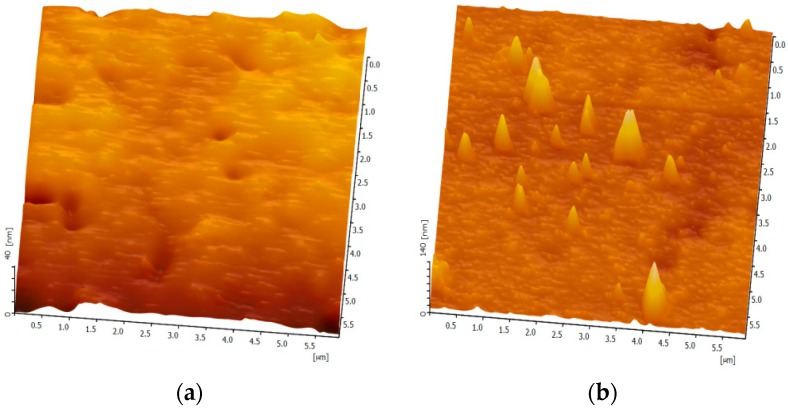
Atomic force microscopy (AFM) morphologies of flat silicon surfaces grafted with PSVBA brushes: (**a**) 20-h grafted surface; (**b**) 48-h grafted surface.

**Figure 5 polymers-11-00981-f005:**
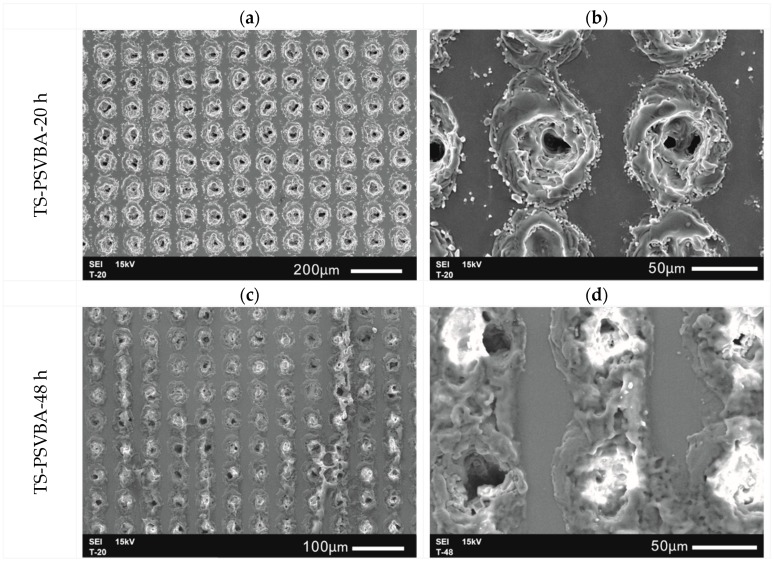
SEM morphologies of textured silicon surfaces grafted with PSVBA brushes: (**a**,**b**) 20-h grafted surface; (**c**,**d**) 48-h grafted surface.

**Figure 6 polymers-11-00981-f006:**
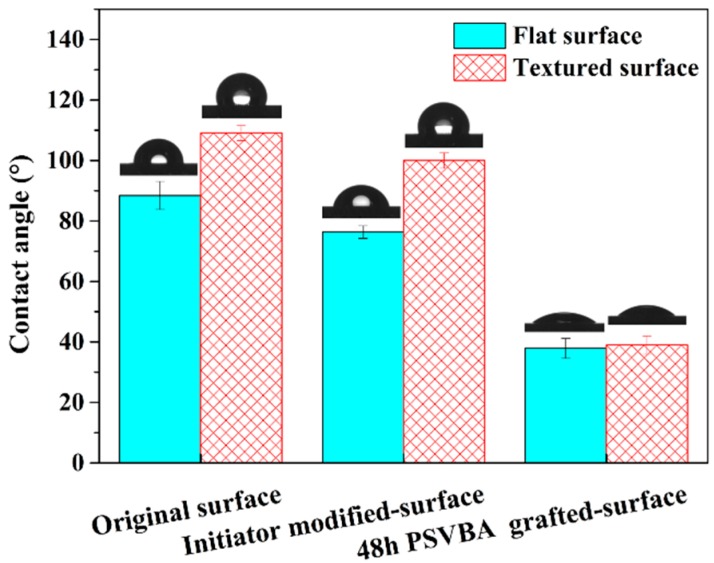
Static contact angles of Si surfaces in saturated NaCl solution with different modifications.

**Figure 7 polymers-11-00981-f007:**
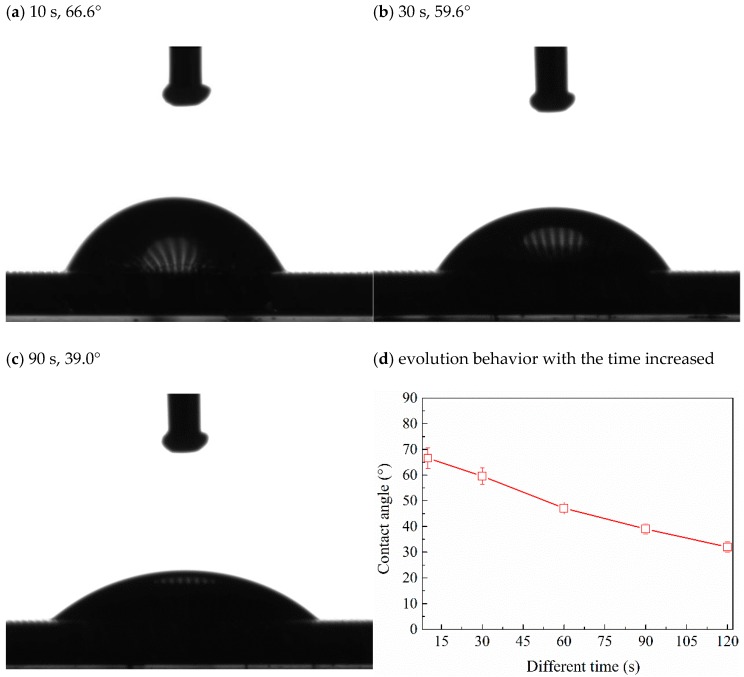
Static contact angle of PSVBA-grafted textured surface at different times (**a**–**c**), and its evolution with time increased (**d**) under saturated NaCl solution conditions.

**Figure 8 polymers-11-00981-f008:**
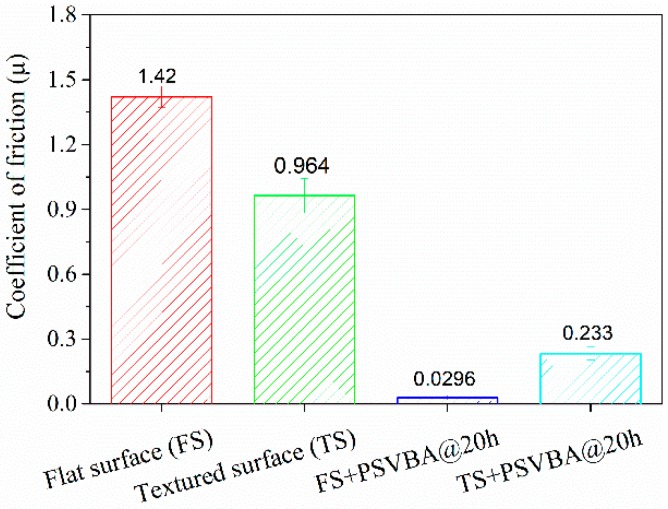
Average friction coefficients at the initial stable stage for different modifications.

**Figure 9 polymers-11-00981-f009:**
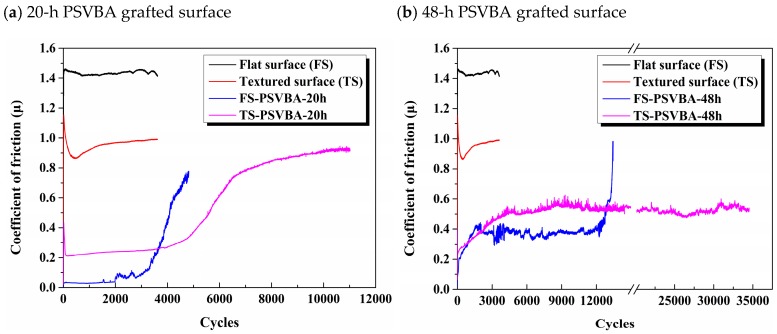
Time-varying curve of friction coefficients of the original and modified surface with different grafting time.

**Figure 10 polymers-11-00981-f010:**
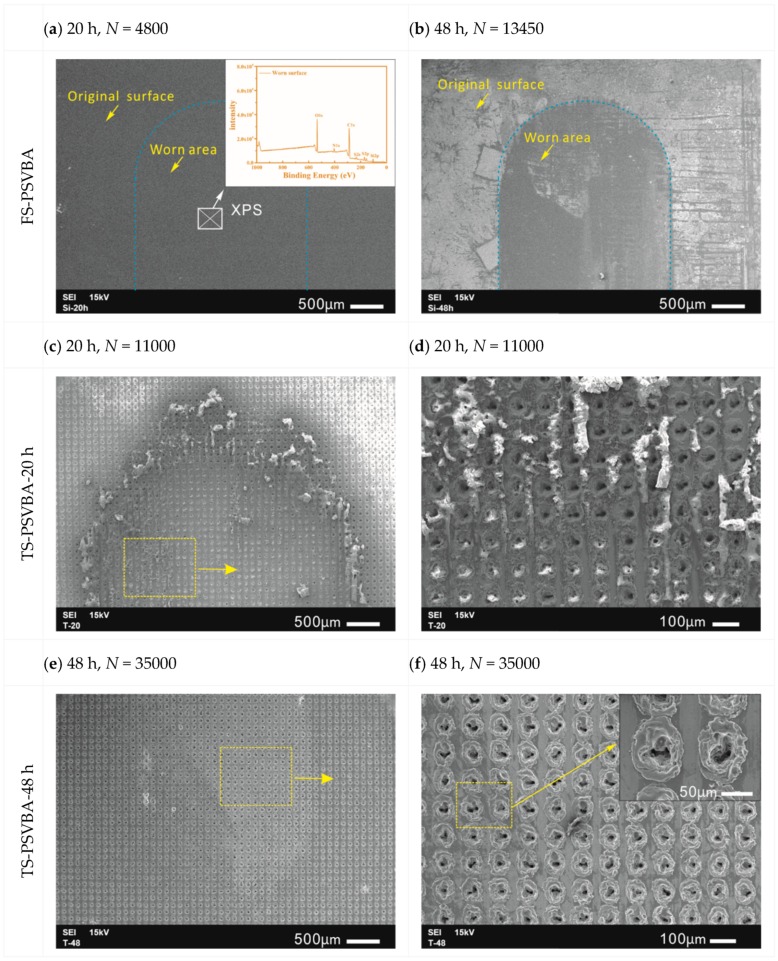
Wear morphology of silicon surface with PSVBA brushes grafted on flat (**a**,**b**), and textured (**c**–**f**) surfaces.

**Figure 11 polymers-11-00981-f011:**
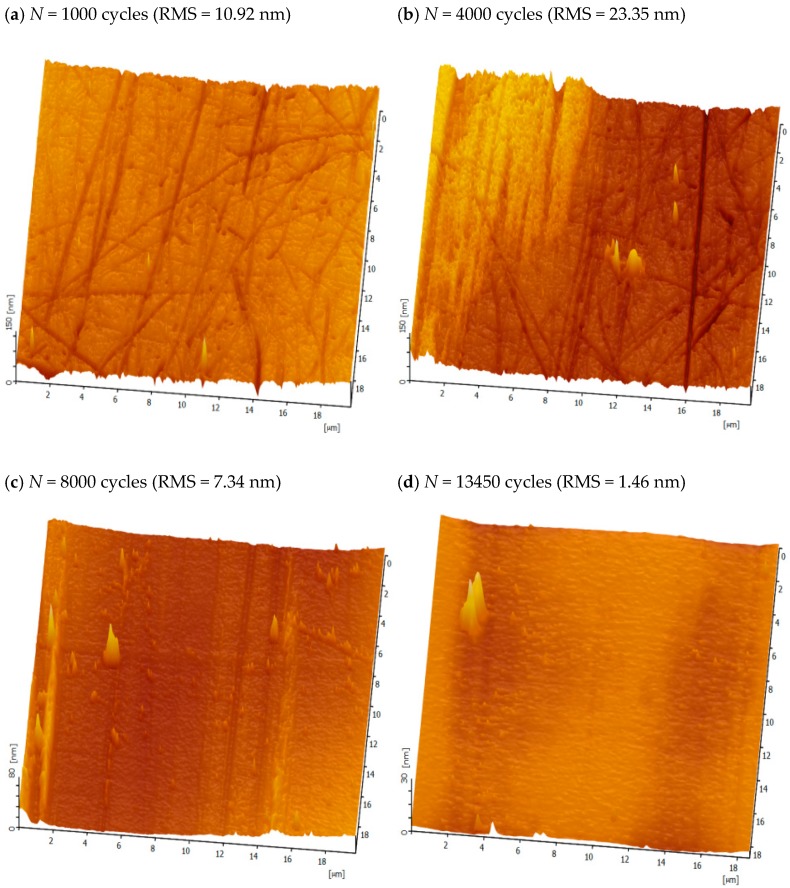
AFM morphology of 48-h PSVBA brushes grafted on flat silicon wafers at different wear stages. RMS—root-mean-square roughness.

**Figure 12 polymers-11-00981-f012:**
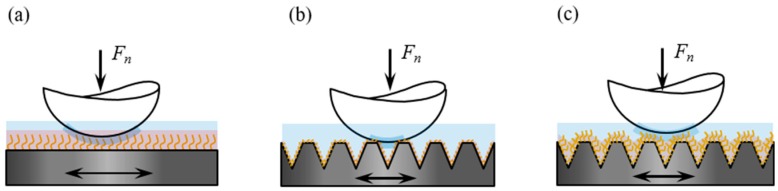
Schematics of the contact mechanics for (**a**) polymer brushes grafted on a flat silicon textured surface, (**b**) thin polymer brushes on a textured surface, and (**c**) thicker polymer brushes on a textured surface.

**Table 1 polymers-11-00981-t001:** Contact angle of experimental results (*θ_textured_*) and values calculated from Wenzel model ($\theta_{w}$) and Cassie–Baxter model (*θ_CB_*) for different type surfaces.

Type	Contact Angle (°)			
	$\theta_{flat}$	$\theta_{w}$	$\theta_{CB}$	$\theta_{textured}$
Original	88.5	88.3	100.1	109.1
Initiator modified	76.4	74.4	90.41	100.0
48-h polySVBA grafted	37.9	25.8	64.0	39.0

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
