# Peer review of "Wetting Behavior and Tribological Properties of Polymer Brushes on Laser-Textured Surface"

_polymers, 2019, doi:10.3390/polym11060981_

Round 1

Reviewer 1 Report

- In the first sentence of the introduction, what really means ultra-low friction?

- The language of the manuscript needs to be improved.

- From the introduction, it remains rather unclear what the connection between brushes and LST is? This needs to be worked out in a better way.

- Recently published review articles on surface texturing (2016 and 2017) are missing.

- What is the effect of the combination of LST and solid lubricant? The authors need to explain that.

- The motivation of this work must be presented and explained in a better way. The last paragraph of the introduction needs to be improved.

- No experimental details related to LST are presented…

- Why did the authors select this specific texture parameters? Please explain and verify and cross-correlate with existing research literature.

- Figure 1 and 2 can be combined.

- Why did the authors use salt solution for the contact angle?

- What is the underlying reason for choosing a PDMS ball as a counter body? Please explain the selection of the tribological testing parameters…

- What do the authors mean by “multistep-modified surfaces”?

- The XPS analysis is rather poor. The respective fits should be presented as well as a deeper interpretation of the results.

- In the SEM micrographs in Figure 6 please delete the magnification and improve the scale bars.

- Instead of presenting individual figures such as in Figure 8, it would be recommendable to present an evolution over time.

- The theory and equations related to Wenzel and Cassie Baxter can be deleted since this is common sense.

- How did the authors evaluated the stable COF? The authors must present error bars for all friction and wear data.

- What is really the underlying mechanism for the observed friction reduction in your samples?

- Please explain the evolution of the COF with time in detail.

- Figure 11: please remove the table. Please delete magnifications and improve the scale bars. 

Author Response

Thank you very much for the valuable and constructive comments, which helped us to significantly improve our manuscript. We have revised our manuscript according to your comments and those from the other reviewers. Details please see the attachment.

Reviewer 2 Report

The article is interesting and well written, however, before the publication, the technological parameters of the laser texturing process should be added, as well as a few references from the last 3 years (there are too few). Images in Fig. 4 should be larger because it is difficult to read values from the axis.

Author Response

(The authors gave the same response as above.)

Reviewer 3 Report

Please read the attched comments.

Author Response

(The authors gave the same response as above.)

Round 2

Reviewer 1 Report

- The language, especially of the newly added parts, must be revised again. - Please specify the used laser wavelength and laser fluency. - The information about the averaging of the COF is not sufficient. Please try to explain that in more detail. - Which kind of background substraction has been used for the XPS fitting?

Author Response

Thanks again for your the valuable and constructive comments, which helped us to significantly improve our manuscript. We have revised our manuscript according to your comments again. Details please see the attachment.

Reviewer 3 Report

The quality of technical paper could be improved enough to be published.

One point to be checked.

In line 81; It is think that → it is thought that ?

Author Response

(The authors gave the same response as above.)

Round 3

Reviewer 1 Report

No further comments.